# Quasi-monoenergetic deuteron acceleration via boosted coulomb explosion by reflected picosecond laser pulse

Tianyun Wei [1] ✉, Zechen Lan [2], Yasunobu Arikawa[2], Yanjun Gu [3], Takehito Hayakawa[1], Alessio Morace[2], Ryuya Yamada[2], Kohei Yamanoi [2], Koichi Honda[2], Masaki Kando [1], Nobuhiko Nakanii[1], Seyed Reza Mirfayzi [4], Sergei Vladimirovich Bulanov [1,5] & Akifumi Yogo [2]

Generation of quasi-monoenergetic ions by intense laser is one of long-standing goals in laser-plasma physics. However, existing laser-driven ion acceleration schemes often produce broad energy spectra and limited control over ion species. Here we propose the acceleration mechanism, boosted Coulomb explosion, initiated by a standing wave, which is formed in a pre-expanded plasma by the interference between a continuously incoming main laser pulse and the pulse reflected by a solid target, where the pre-expanded plasma is formed from a thin layer on the solid target by a relatively strong pre-pulse. This mechanism produces a persistent Coulomb field on the target front side with field strengths on the order of TV/m for picoseconds. We experimentally demonstrate generation of quasi-monoenergetic deuterons up to 50 MeV using an in-situ $D_2O$-deposited target. Our results show that the peak energy can be tuned by the laser pulse duration.

The ability of producing ion sources with high brightness using ultra-intense lasers[1,2] has provided a platform to develop various applications such as compact neutron source[3–6], fast ignitions[7–10], proton radiography[11–13] and radiobiological research[14–16]. Protons with energies of 150 MeV have already been realized[17], which shows the possibilities of further applications such as cancer therapy[18,19]. Many ion acceleration mechanisms have been proposed, including Collisionless Shock Acceleration (CSA)[20,21], Radiation Pressure Acceleration (RPA)[22,23], and Target Normal Sheath Acceleration (TNSA)[24,25]. Among these mechanisms, TNSA is widely applied in most laser systems. CSA and RPA are considered to be more efficient and reliable mechanisms rather than TNSA, but the requirement for the target thickness and laser intensity is limiting their application in current laser systems.

There are two remarkable features of the TNSA mechanism. The first is that ions in a contamination layer on a target surface are first accelerated, mainly making it difficult to accelerate effectively other types of ions, rather than protons. Thus, previous studies have proposed some approaches to accelerate other types of ions in TNSA, such as using a cryogenic $D_2$ target to realize pure deuteron acceleration[26], heating the target surface by additional laser for heavy ion acceleration[27], and focusing an ultra-intense laser on a nanometer-scale target[28]. The second is that the accelerated ions typically have a continuous distribution, which is not suitable for some applications, such as the study of nuclear physics and nuclear engineering. For example, monoenergetic deuterons via high-power laser can be used for neutron sources with selective energy using the secondary reactions such as the $d+{}^9Be$ reaction,[29] the study of $d+d$ and $d+t$ nuclear fusion[30,31], and production of medical radioisotopes[32]

Producing quasi-monoenergetic particles using lasers[33] remains a significant challenge. Many efforts[34–45] including different acceleration mechanisms, have been made to generate quasi-monoenergetic ions, for example, using the ultra-thin target[34–36], the microstructured

[1]Kansai Institute for Photon Science (KPSI), National Institutes for Quantum Science and Technology (QST), Kizugawa, Kyoto, Japan. [2]Institute of Laser Engineering, The University of Osaka, Suita, Osaka, Japan. [3]Institute of Scientific and Industrial Research (SANKEN), The University of Osaka, Ibaraki, Osaka, Japan. [4]Tokamak Energy ltd, Milton, United Kingdom. [5]Extreme Light Infrastructure ERIC ELI Beamlines Facility, Dolní Br^ežany, Czech Republic. ✉e-mail: wei.tianyun@qst.go.jp

target[18,37–39] and the coil target[40]. The simplest method is to use the ultra-thin target. When the thickness of the thin layer is thin enough to be collectively affected by the laser field, quasi-monoenergetic ions could be accelerated.

In our recent study[45], we demonstrated a method for the fabrication of in-situ $D_2O$ deposited targets, achieving acceleration of quasi-monoenergetic deuterons ($\Delta E/E = 4.6\%$) through the TNSA mechanism. However, protons play significant roles in the formation of the quasi-monoenergetic component of deuterons. The electric field generated by the accelerated protons suppress the energies of the high-energy deuterons to form quasi-monoenergy deuterons. This suppression limits the peak energy of the deuterons to approximately 11 MeV. When we use a target with a proton layer, it is difficult to accelerate deuterons much higher than 11 MeV. Thus, we need an alternative mechanism to accelerate quasi-monoenergetic high-energy deuterons.

Coulomb explosion[41–44] is another possible mechanism to generate quasi-monoenergetic ions, which has the potentiality to generate ions with a peak energy of several tens of MeV in the backward direction of the laser[46]. A combination of the charge separation electric field and the field of the laser can result in the generation of quasi-monoenergetic proton beams in so-called the "directed Coulomb explosion acceleration regime"[43]. However, because of the low efficiency of the Coulomb explosion, the energies of accelerated ions are limited to only 1 ~ 2 MeV in the previous experiments[41,42]. In addition, it has been theoretically proposed that protons could be accelerated up to 100 MeV through a longitudinal charge-separation field generated by chirped standing waves formed by a fs laser pulse reflected by a high-density mirror located behind a target[47].

In this study, we investigate that a standing wave generated by reflecting a picosecond laser pulse with the critical surface of a plasma[48] could enhance the efficiency of the Coulomb explosion for producing a persistent electric field. We conduct an ion acceleration experiment using Al targets with ultra-thin $D_2O$ layer targets using the in-situ fabricated method. We introduce a pre-pulse to expand the ultra-thin $D_2O$ layer before the laser main pulse. As a result, we measured high-energy quasi-monoenergetic deuterons with energies of up to 50 MeV. Simulation results reveal the mechanism that dominates the acceleration of the quasi-monochromatic deuterons. A relatively

strong pre-pulse expands a thin $D_2O$ layer on a solid target to form an underdense plasma, and subsequently, a picosecond main laser pulse can penetrate through this plasma but is reflected by the solid density region. The reflected pulse and the continuously incident laser pulse then form a standing wave at the front of the target, which effectively kicks out electrons and generates a strong Coulomb explosion field. This field, on the order of TV/m, persists for several picoseconds and enables the acceleration of deuterons to quasi-monoenergetic spectra with peak energies reaching tens of MeV.

## Results

The typical raw data of the Thomson Parabola Ion Spectrometers (TPISs)[49,50] for the shots with only the main pulse are shown in Fig. 1a, b. There are low-energy deuterons measured only at the laser-facing side of the target. In contrast, the typical raw data of the TPISs for the shots with pre-pulse are shown in Fig. 1c, d. Deuterons are still only measured at the laser-facing side, where the peak energy of the deuterons is approximately 40 MeV. The energy spectra of 3 different shots with a pre-pulse at the laser-facing side are shown in Fig. 1e–g. Each energy spectrum is obtained from a single laser shot without accumulation. The peak energies vary from 20 MeV to 50 MeV. The shots L5158 and L5168 have similar laser parameters (see Table 1), including pulse durations and energies. This is further supported by the similar proton spectra in Fig. 2. In contrast, they exhibit different deuteron spectra as shown in Fig. 1e, g. This variation may arise from the difference in the fabrication condition of the $D_2O$ layers or the reproducibility of the pre-plasma generation. The present result indicates that we may control the deuteron peak energy by optimizing the thickness of the $D_2O$ layer and the laser parameters. Note that the TPISs used in the experiments have an energy resolution better than 0.2 MeV in the 3–30 MeV range. At higher energies (> 30 MeV), the resolution decreases [see the dashed lines in the TPIS raw data in Fig. 1a–d], which may broaden the observed spectral features.

The spatial pattern of the highest energy deuterons measured by the Radiochromic Films (RCF)[51] in the beam profile shot is shown in Fig. 1h. By calculating the particle transport with the PHITS Monte Carlo simulation code[52], the minimum energies of the protons and deuterons reaching the RCF layer are presented in Figs. 1h, i through other layers is 24.6 MeV and 33.0 MeV, respectively. Because the

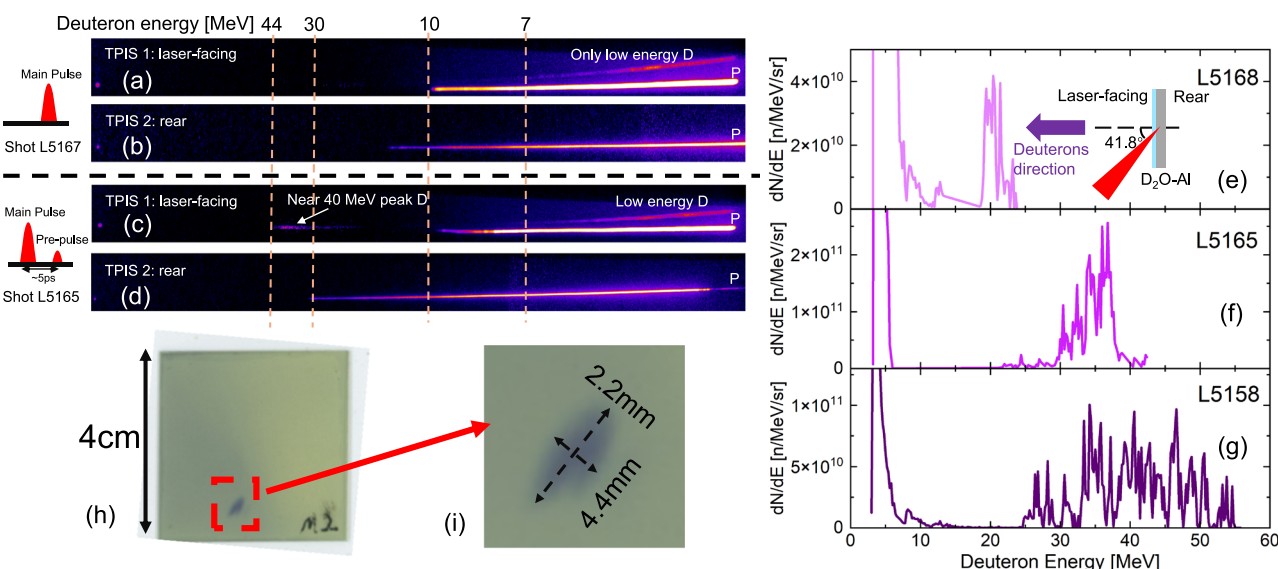

**Fig. 1 | Experimental results. a, b** Raw data of TPISs for shot L5167. Only low-energy deuterons are measured on the target laser-facing side. **c, d** Raw data of TPISs for shot L5165. Shot with pre-pulse, ~40 MeV quasi-monoenergetic deuterons are measured at the laser-facing side. **e–g** The deuteron spectra at the laser-facing side for 3 different shots. Structured deuterons from 20 to 50 MeV are measured. **h** The measured highest energy profile by RCF, the corresponding deuteron energy is 33.0 MeV. **i** The spot is nearly an ellipse with a size of 4.4 mm × 2.2 mm.

**Table 1 | The shot-by-shot laser energy and pulse duration**

| Shot No. | Pre-pulse | Pulse FWHM (ps) | | | | Laser energy on target (J) | | | | | D⁺ peak (MeV) |
|---|---|---|---|---|---|---|---|---|---|---|---|
| | | H1 | H2 | H3 | H4 | H1 | H2 | H3 | H4 | Total | |
| L5158 | w | 1.89 | 1.80 | 1.53 | 1.73 | 160 | 163 | 162 | 118 | 603 | 40.6 |
| L5165 | w | 1.92 | 1.78 | 1.83 | 1.68 | 182 | 190 | 189 | 138 | 699 | 34.6 |
| L5167 | w/o | 1.87 | 1.74 | 1.44 | 1.60 | 167 | 181 | 167 | 133 | 648 | – |
| L5168 | w | 1.95 | 1.82 | 1.68 | 1.66 | 173 | 175 | 175 | 128 | 651 | 20.4 |
| L5169 | w | 1.72 | 1.60 | 1.53 | 1.44 | 192 | 194 | 188 | 143 | 717 | 33.0 |
| L5171 | w/o | 1.74 | 1.61 | 1.59 | 1.44 | 182 | 184 | 181 | 134 | 681 | – |

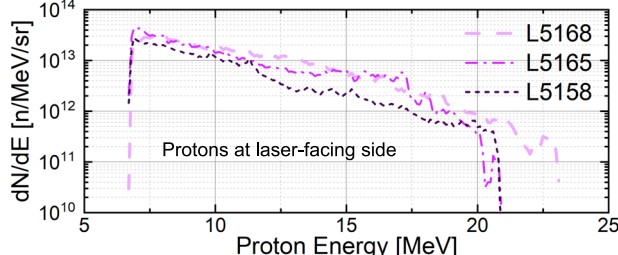

**Fig. 2 | Proton spectra.** The proton spectra at the laser-facing side for the same 3 shots with TPIS.

maximum energy of the protons is 23 MeV, as shown in Fig. 2, the signals in the RCF originate only from the deuterons. It is difficult to evaluate the deuteron energy spectrum lower than 33.0 MeV from the RCF stacks, because both protons and deuterons make signals in the lower energy RCF layers. Because the laser irradiates the target at an angle from the normal direction of the surface, the deuteron profile shows a nearly elliptical spot with a size of 4.4 mm × 2.2 mm. By considering the distance of the RCF from the target (31 mm), the divergence of the high-energy deuterons is estimated as $\Omega = \pi \times 3.3 \times 4.4/31^2 \approx 0.03$ sr. Assuming an axially symmetric conical distribution, the half angle $\theta \approx 5.6°$ is derived from $\Omega = 2\pi(1 - cos\theta)$.

The deuterons are accelerated from the laser-facing side of the target, and their energies are higher than those of the simultaneously accelerated protons. These facts indicate that the quasi-monoenergic deuterons are accelerated by a mechanism different from the TNSA in the previous study[45]. Note that the detection threshold of the TPIS is approximately 3 MeV for deuterons.

To understand the acceleration process of the structured deuterons at the laser-facing side of the target, we conduct 2D Particle-In-Cell (PIC) simulations with the EPOCH code[53] using the set-up shown in Fig. 3a. The energy spectra of the deuterons at the laser-facing side of the target in a time range from t = 2.04 ps to t = 4.44 ps are shown in Fig. 3b. The energy spectra measured in the shot L5165 and L5158 are also included in Fig. 2b to illustrate the agreement between simulation and experiment results. The deuterons show a structured component around 15 MeV, which appears at 2.04 ps. The energy of this component is further increased and becomes stable at 3 ps. Fig. 3c shows the evolution of the deuteron peak, where its energy starts to increase after 1 ps, and rapidly increases up to 2.7 ps. It finally became stable after 3 ps with a peak energy of approximately 35 MeV. Fig. 3d shows the deuteron peak energy dependence on the pulse duration of the incident laser. The pulse durations are changed from 0.8 ps to 1.8 ps while the incident laser energies remain constant by decreasing the laser intensity. The bandwidth of the peak components is adopted as the error bar for the peak energy. The calculated result [Fig. 3d] shows that the maximum peak energy of the accelerated deuterons is obtained in the case of a pulse duration of 1.5 ps, which is nearly equal to the typical pulse width of the LFEX laser. If we use laser pulses longer

than 1.5 ps, the peak intensity and normalized vector potential $a_0$ decrease ($a_0 \propto 1/\sqrt{\tau}$) under the condition that the total laser pulse energy is fixed. As $a_0$ decreases, the penetration depth of a laser pulse in a relativistic plasma (relativistic skin depth) shortens, and the amplitude of the reflected laser to form a standing wave decreases. These effects reduce the strength of the boosted Coulomb explosion field to accelerate deuterons. As shown in Fig. 3d, when pulse widths are longer than 1.5 ps, the peak energies of the accelerated deuterons are lower than that at 1.5 ps by approximately 5–8 MeV. The energy spectrum of the accelerated deuterons depends on the density and the scale length of the pre-plasma. In the present condition, the pre-expanded $D_2O$ layer has an exponential profile with a few-micrometer scale length. If the pre-plasma has a thin scale length or low density, a long laser pulse penetrates easily the pre-plasma and accelerates effectively deuterons. Conversely, when the pre-plasma has a thick scale length or a high density, a shorter laser pulse is suitable for effective deuteron acceleration.

The electron density while the laser main pulse incident on the solid target (1.08ps) is shown in Fig. 4a. The electrons are heated and transparented into the target rear side, forming the sheath field at the target rear. In contrast, at the laser-facing side, electrons are quickly kicked out by the laser pulse. As a result, a strong Coulomb explosion field is formed at the target front side by the left positive charged ions. The Fig. 4b–d shows the 2D distribution of $B_z$, indicating the position of the laser pulse. The time stamps in these figures show the simulation time. The laser pulse is set from the left boundary (−200 μm) with a pulse width of 1.5 ps and a peak at 1 ps. This pulse reaches the surface of the target at approximately 0.67 ps later, and the peak of the pulse reaches this surface at approximately 1.67 ps.

Figure 4e–g shows the 2D energy distribution of the deuterons, and the black dashed lines show the spatial mean value of $E_x$ field at the center (-2.5 μm to 2.5 μm). At an early time (2.04 ps) the laser incident on the solid target, and the laser pulse is reflected by the solid target. The reflected pulse and the continuous incident pulse form a standing wave that kicks out the electrons effectively to lead the Coulomb explosion field at the laser-facing side. The deuterons at the laser-facing side are rapidly accelerated by this field. At a later time (2.64 ps), the incidence of the laser is already finished, but the reflected laser pulse kicks out the electrons and maintains the Coulomb explosion field. This field exists just behind the highest energy deuterons and accelerates deuterons during a sub-picosecond duration. Once the laser, including the reflected laser, has completely disappeared at 3.24 ps, the electrons expand to the laser-facing side, and the process is terminated, and the deuteron energy is becoming stable. The Coulomb explosion field maintains its strength in the order of ~TV/m for multi-picoseconds, after which the laser incident on the solid target because the main and reflected laser pulses kick out the electrons in multi-picoseconds. The electric field generated by electrons at the laser-facing side could be estimated from the electron density distribution shown in Fig. 4a. The estimated field is on the order of $10^9$ V/m, which is much lower than the Coulomb explosion field strength in the order of TV/m. This Coulomb explosion field accelerates the deuterons to the

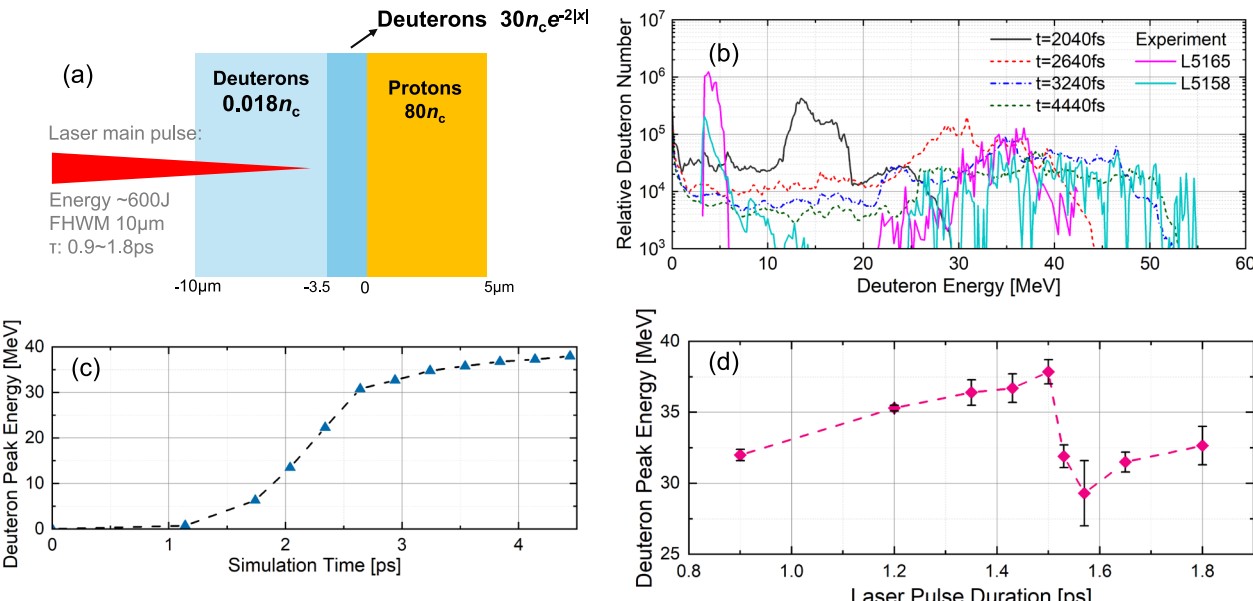

**Fig. 3 | Simulation set-up and deuteron energy evolution. a** Simulation set-up, self-focused main pulse incident into the solid target with pre-expanded deuterons. **b** Energy spectra at different simulation times. Experimental spectra of L5165 and L5158 are also included. **c** The deuteron peak energy time evolution. **d** Laser pulse duration dependence of the deuteron peak energy.

maximum energy of ~ 50 MeV. The acceleration field exits behind the highest energy deuterons, and there are no electrons that form other electric field to expand the deuteron energy so that the deuterons keep the peak energy during the acceleration phase. In addition, the accelerated deuterons have an arc shape, which is formed by the hole boring effect[54] of the laser main pulse. This allows the highest energy deuterons to collimate themselves around the center along the target normal direction. This is consistent with the measured small divergence.

Note that multi-peak structures are observed in all three experimental shots [Fig. 1e–g]. During the early stage when the standing wave forms at tens of femtoseconds, electron modulation may induce a multi-peak structure on the deuterons[47] before the main Coulomb explosion acceleration.

In the above simulations, the angle of the incident laser has been assumed to be 0° from the normal direction of the target surface. Because the incident angle in the experiment is 41.8°, we perform an additional simulation where the incident angle of 42° is assumed. Figure 5a presents that a standing wave still forms in the overlap region of the incident and reflected pulses, and the quasi-monoenergetic deuterons are accelerated up to approximately 80 MeV along the normal direction with an open divergence angle of smaller than 24° shown in Fig. 5b. The standing wave leads to localized electron evacuation, and thus a strong electrostatic field is generated along the normal direction. This accelerates deuterons preferentially along the normal direction. The divergence angle obtained in the simulation calculation is wider than that of the experimental result. This may arise from the idealized assumptions used in the simulation, particularly the simplified target density profile. It is considered that in the experiment, the front surface of the pre-expanded target becomes a Gaussian like surface instead of the flat surface. In addition, the simulation is performed in 2D geometry, which may lead to a different, typically wider, divergence compared with that of 3D simulations.

The simulation result shows clearly that the boosted Coulomb explosion mechanism dominates the acceleration of the deuterons in the quasi-monoenergetic peak at the laser-facing side. There are two key points for this mechanism. The first is that a thin pre-plsama should exsit on the laser-facing side of a target, allows the incident laser has a large skin-depth to kick the electrons out of the target to create a strong Coulomb explosion field in the order of TV/m. In our experiments, to make such thin pre-plasma, we deposited target in-situ. This method creates a ultra-thin $D_2O$ layer which could be pre-expanded to a thin plasma by a pre-pulse with a fraction of ~ 5% of the main pulse. The second is that a relatively longer laser pulse to maintain the field in a multi-picoseconds. The LFEX laser has a pulse duration of ~ 1.5 ps, which allows the acceleration field could last to ~ 2 ps because the reflect laser could also kick out the electrons to maintain the Coulomb Explosion field. This method could apply to other picosecond lasers, such as OMEGA EP[55] and PETAL[56], for quasi-monoenergetic ion acceleration under energy control.

Finally, we discuss a possibility that we form a boosted Coulomb explosion to other ion species such as tritons and protons. The accelerated ions could be selected by depositing a contamination layer on a metal target. We use a $D_2O$ deposited target for deuteron acceleration; it is acceptable to realize quasi-monoenergetic triton acceleration, which is expected in research such as nuclear fusion, avoiding the possible high cost of isotopes and radiation hazards in traditional accelerators[31]. As for quasi-monoenergetic proton acceleration, which is necessary for applications such as cancer therapy, because it is difficult to make an ultra-thin layer of protons on a metal target, we suggest the use of a stronger pre-pulse to expand quickly the proton plasma.

In this work, we investigated the boosted Coulomb explosion mechanism as a means of accelerating ions to energies of several tens of MeV using picosecond laser pulses. Through this mechanism, we experimentally achieved quasi-monoenergetic deuterons with peak energies up to 50 MeV by irradiating in-situ $D_2O$-deposited targets with the LFEX laser system. A thin $D_2O$ layer was formed on an aluminum target by the evaporation of $D_2O$ molecules through nanoscale holes in a plastic capsule containing liquid $D_2O$. This layer was pre-expanded by a pre-pulse, enabling the main laser pulse to transmit through the underdense plasma and reflect off the solid aluminum surface. Ion acceleration at the laser-facing side is initiated by a standing wave formed by the interference between the reflected and continuously incident picosecond laser pulses. This standing wave efficiently expels electrons, generating a strong Coulomb explosion field on the order of

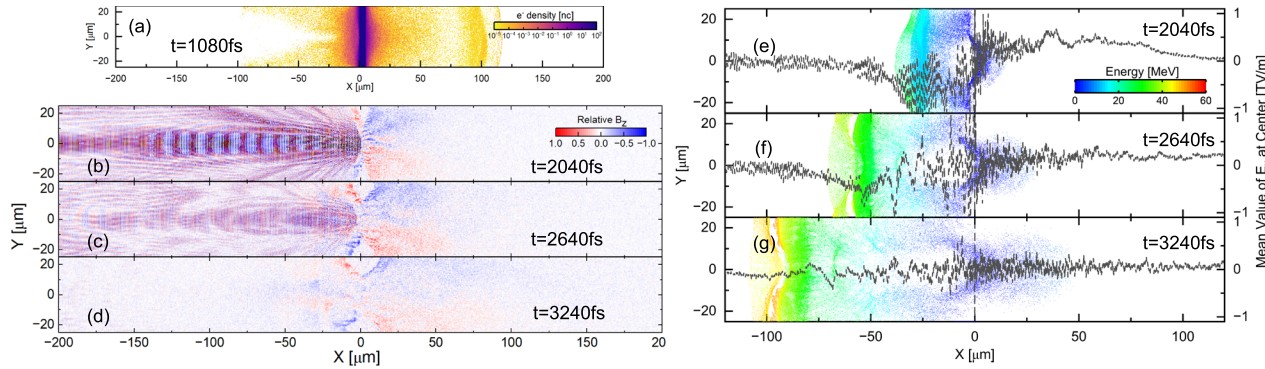

**Fig. 4 | Simulated dynamics of the deuteron acceleration process. a** Electron density while laser incident on the solid target. Electrons are kicked off by the laser pulse at the laser-facing side of the target. **b–d** The $B_z$ field indicates the laser pulse position during simulation. **e–g**The deuteron energy distribution and $E_x$ field at the center, deuterons are accelerated by a TV/m level Coulomb Explosion field lasting multi-picoseconds.

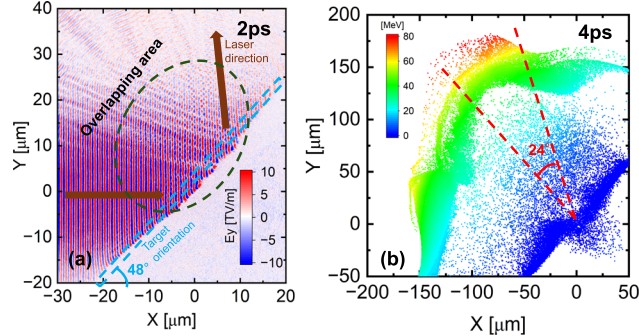

**Fig. 5 | Simulations for oblique incidence. a** Standing wave is formed at the overlapping area of the incident and reflected pulses. **b** Deuterons are able to be accelerated into several tens of MeV on the laser-facing side at normal direction.

TV/m, persisting for several picoseconds. Simulations show that ions can be accelerated to tens of MeV with a narrow energy spread, and that the peak energy can be tuned by adjusting the laser pulse duration. The boosted Coulomb explosion mechanism provides an energy-controllable approach for generating quasi-monoenergetic ions using high-power laser systems. Moreover, the ion species can potentially be selected by depositing appropriate surface layers on the target, offering flexibility in tailoring ion sources. This mechanism holds strong promise for a wide range of applications, including fast ignition in inertial confinement fusion, nuclear physics experiments, and laser-driven cancer therapy.

## Methods
### Experiments
The experiments are conducted with the Laser for Fast Ignition Experiment (LFEX) system[57] at the Institute of Laser Engineering in the University of Osaka. The LFEX laser provides three shots per day and four pulses of H1–H4 with a center wavelength of 1.05 μm are simultaneously delivered for each shot. The focal spot diameters of the four pulses are 50 μm for H1, H3, and H4 and 30 μm for H2. The laser pulse energy and duration of each shot used in the experiments are listed in Table 1. The laser is operated in two modes. The first mode is a normal shot with the main pulse only, and the second mode is a shot with a pre-pulse approximately 5 ps before the main pulse. The energy of the pre-pulse is approximately 5% of the main pulse. For all the shots with pre-pulse, we have high-energy components of deuterons at the laser-facing side, which indicate a high reproducibility.

The experiment set-up is shown in Fig. 6a. The LFEX laser irradiates the $D_2O$-Al target front side with 41.8°, and two TPISs are set on the front side and rear side to identify ion species and measure the energy

spectra of the accelerated ions. The Al filters with a thickness of max 300 μm are used in the TPISs to stop heavy ions such as carbon and oxygen, so that only protons and deuterons are measured with the TIPSs. The RCF and Al filter stack with a size of 4 × 4 cm is set at the front side of the target with a distance of 3.1 cm in the deuteron pulse profile shot after the measurement of the energy spectra with TPISs. The RCF stack was placed parallel to the target surface but offset by approximately 1 cm from the centerline to avoid interference with the laser beam. Each Al filter is inserted between two RCFs to decrease the deuteron energy. A $D_2O$-Al target[45] is in-situ fabricated using a $D_2O$ capsule to deposit a $D_2O$ layer on the surface of an Al target, as shown in Fig. 6c. Before a laser shot, we locate a spherical plastic capsule including $D_2O$ water near an Al target. A part of $D_2O$ molecules evaporate through nano-scale holes of the plastic capsule, and subsequently accumulate on the surface of the Al target so that a thin $D_2O$ layer is formed. The thickness of the $D_2O$ layer is approximately proportional to the deposition time. The depositing times of ~ 40 min are constant for every shot to make every target have almost the same thickness of the $D_2O$ layer. The $D_2O$ capsules are removed from the target chamber before laser shots. The Al targets are kept at near room temperature (25 °C) during the deposition process.

### Simulations
The simulation box is in the region of from -200 to 200 μm for the x-axis and from -25 to 25 μm for the y-axis with a mesh size of 50 nm. As shown in Fig. 3a, protons with a density of 80 $n_c$ are set in the solid Al target region from 0 to 5 μm for the x-axis. The protons are used instead of Al ions because of computational efficiency. By considering a laser pre-pulse, low-density deuteron plasma with a density of 0.018 $n_c$ is set in the region from −10 to −3.5 μm for the x-axis to present the pre-pulse expansion, which is estimated to have a temperature of 10 keV over 5 ps. To represent the effect of the pre-pulse, which is 5 ps before the main pulse in the experiment, we estimate the most expanded size of the low-density plasma in front of the target by assuming a deuteron temperature of ~ 10 keV. The ion sound speed corresponding to this temperature gives an expansion distance of approximately 5 μm within 5 ps. Note that the acceleration process is not sensitive to this deuteron layer because of its low density. High-density deuterons is set in the region from −3.5 to 0 μm for the x-axis to represent the fully ionized $D_2O$ layer (~ 33$n_c$). This deuteron layer has an exponential distribution of 30 $n_c e^{-2|x|}$ at the laser-facing side of the proton layer. Note that oxygen ions are not set in the deuteron plasma because of computational efficiency. A laser pulse with a total energy of 600 J (typical total energy of LFEX) incidents into the target from the thin deuteron plasma side. The laser pulse duration is 1.5 ps in FWHM in the simulations. The wavelength is 1.05 μm identical to the experimental parameter. The laser spot size is set to 10 μm in FWHM instead

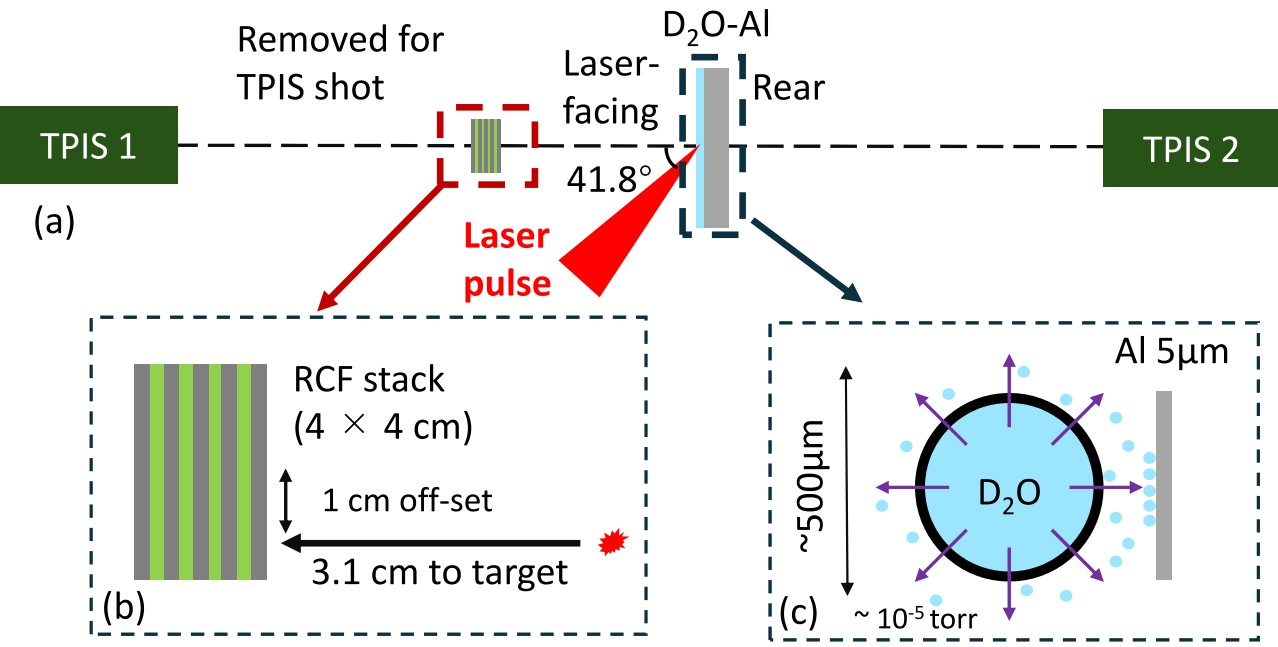

**Fig. 6 | Experimental set-up. a** Deuteron acceleration experiment set-up with the D$_2$O-Al target using the LFEX laser. **b** RCF stack set 3.1 cm at the front of the target in deuteron beam profile shot. **c** D$_2$O-Al target in situ fabricated using a D$_2$O capsule in the vacuum chamber ( ~ $10^{-5}$ torr).

of the size in the experiments of 50 μm to account for self-focusing effects and to reduce computational complexity. The laser intensity of the simulation increases from the experiment intensity of $10^{19}$ W/cm$^2$ to approximately $2.5 \times 10^{20}$ W/cm$^2$. This higher intensity allows deeper penetration of the main pulse into the pre-plasma, but the reflected surface is shifted to a position of a higher electron density of $n_c\sqrt{1+a_0^2/2}$, which is approximately 4.5 times higher than that in the experiments. Since the focal spot size is 5 times smaller in the horizontal direction, the total number of evacuated electrons remains comparable to the experimental case. As a result, the total number of evacuated electrons in the simulation is comparable to that in the experiments. This leads that the present simulation giving the Coulomb explosion field strength similar to the experimental condition.

The simulation box for the oblique incidence in Fig. 5 is in the region of from −200 to 50 μm for the x-axis and from −100 to 300 μm for the y-axis with a mesh size of 50 nm. The laser spot (FWHM) is set as 25 μm with the same laser energy. The target set-up is the same with that for the normal incidence.

## Data availability

The experimental raw data and calculated energy spectra supporting this study are publicly available in the figshare dataset[58]. Additional support for the data can be requested from the corresponding author, who will respond within one working week.

## code availability

The input files for simulations are publicly available in the figshare dataset[58]. The full simulation generated files are extremely large (multi-terabyte scale) and cannot be deposited in a public repository; therefore, they are available from the corresponding author upon request, who will respond within one working week. Once access is granted, the data will remain available without time limitation.

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

## Acknowledgments

The authors thank the technical support staff of ILE for their assistance with laser operation, target fabrication, and plasma diagnostics. T.W. was supported by JST SPRING (JPMJSP2138) and the University of Osaka Honors Program for Graduate Schools in Science, Engineering and Informatics during his PhD studies before September 2024. R.Y. is supported by JST SPRING (JPMJSP2138) and the University of Osaka Honors Program for Graduate Schools in Science, Engineering and Informatics. T.W. acknowledges the use of the HPE SGI8600 supercomputer at the National Institute for Quantum Science and Technology (QST). Y. G. acknowledges the use of the large-scale computer systems at D3 center, The University of Osaka. Portions of this research were carried out by S.V.B. at the ELI Beamlines Facility, a European user facility operated by the Extreme Light Infrastructure ERIC. The authors are especially indebted to Prof. Kunioki Mima, who passed away in January 2025, for his invaluable initial discussions on the theoretical aspects.

## Author contributions

T. W., Z. L., Y. A., T. H., A. M. and R. Y. performed the experiments. T. W. and Z. L. analyzed the experimental data. T. W. conducted the simulations. Y. G. discussed the theories and gave advice on the simulations. K. Y. and K. H. fabricated the capsule targets. M. K. and N. N. discussed the theories. S. R. M. gave the original ideas of using heavy water and reviewed the paper. S. V. B. reviewed the paper from theoretical viewpoints. The manuscript was prepared by T. W. and T. H. All authors contributed to discussions and the preparation of the manuscript. A. Y. provided overall supervision of the work.

## Funding

A.Y. discloses support for the research of this work from JSPS KAKENHI [grant numbers JP25420911, JP26246043, JP22H02007], JST A-STEP [grant number AS2721002c], and JST PRESTO [grant number JPMJPR15PD]. T.W. discloses support for the research of this work from the Collaboration Research Program of ILE, The University of Osaka [grant number 2023A1-004WEI] and JSPS KAKENHI [grant number JP25K23369]. T.H. discloses support for the research of this work from JSPS KAKENHI [grant number JP22H01239]. M.K. discloses support for the research and publication of this work from JSPS KAKENHI [grant number JP23K20038]. The other authors declare no relevant funding.

## Competing interests

The authors declare no competing interests.
