## [Transparent Peer Review file · Nature Communications]

Quasi-monoenergetic Deuteron Acceleration via Boosted Coulomb Explosion by Reflected Picosecond Laser Pulse

Corresponding Author: Dr Tianyun Wei

Version 0:

Reviewer comments:

Reviewer #1

(Remarks to the Author)

Quasi-monoenergetic Deuteron Acceleration via Boosted Coulomb Explosion by Reflected Picosecond Laser Pulse

Summary, noteworthy results and significance

The authors describe their experiment of directing a short-pulse laser at an angle onto a D₂O-covered aluminum target. The laser-facing side emitted deuterons with mean energies between 20 and 40 MeV and widths between ~2 and 15 MeV, a significant improvement over typical, exponential laser ion acceleration spectra. The authors explain this atypical, quasi-monoenergetic spectrum by a new laser-ion acceleration mechanism, a 'boosted Coulomb explosion' caused by the reflected laser beam. Simulations from a corresponding 2D PIC simulation are presented to support this conclusion. The work is an extension of the author's previous study (Wei, T. et al, *Physics of Plasmas* 31(7), 073903 (2024)). The main improvement is an extension of their results to higher deuteron beam energies by accelerating deuterons off the front (laser-facing) side instead of the back. Other studies have limited themselves to deuteron energies <10 MeV, whereas this study went up to 40 MeV, albeit with larger energy spread. Therefore, it is a valuable addition to the body of literature.

Specific comments

- On p.3, it is difficult to follow what 'rear' and 'front' side mean and where the D₂O layer is. The authors may consider using 'laser-facing' instead.
- The figure 1 (a)-(e) labels seem shifted in the text
- The specific laser parameters should be included for each shot shown in figure 1 (e)-(g)
- The authors present RCF to show the spatial beam profile. It is not clear from the text whether the RCF is also sensitive to protons, which could obscure the deuteron beam profile
- Figure 2 (b) would benefit from including the experimental deuteron spectrum to show a match between simulation and experiment
- Figure 2 reveals that the simulation used a laser pulse impinging on the target at 90°. However, the experiment shown in figure 4 shows a 41.8° incidence. The TNSA mechanism is generally robust with respect to this angle; however, the authors later argue for the generation of a standing wave between incident and reflecting laser. This would require the wave vectors sum to zero, which is unlikely at this angle. The authors should repeat their PIC simulation with this angle, or otherwise demonstrate that a standing wave can form under these conditions.
- The simulation includes densities for the solid target and deuteron plasma, which should be explained or justified
- On p.4, the authors make the interesting observation that for pulse durations >1.5 ps, the reduced laser intensity cannot penetrate the ionized deuterons. How does this depend on the chosen densities, and are there other reasonable choices?
- On p.5, the authors argue that the high intensity laser pulse ejects most electrons at the laser-facing side, leaving the ions, which then primarily interact with themselves, leading to a Coulomb explosion that produces the quasi-monoenergetic deuterons. The argument would be significantly strengthened if the authors could separately calculate the electric field generated by the electron density shown in figure 3 (a) and demonstrate that it is insignificant.
- On p.7-8, some more experimental details should be given, in particular the temperature of the Al target during deposition and the specific laser parameters used.

Reviewer #2

(Remarks to the Author)

The manuscript by Wei et al, presents some interesting observations of narrow band deuterium spectra from LFEX irradiations of Deuterium-coated substrates.

The narrow band, high energy features, observed in the target front direction appear only when a prepulse is added to the interaction, suggesting a mechanism taking place in a pre-expanded deuterium-containing plasma. The author suggest a mechanism based on Coulomb explosion in a region of the front plasma where the electrons are evacuated by a standing wave arising from interference of the incident laser and the reflected laser. The interpretation is supported by 2D-PIC simulation showing a similar acceleration pattern.

Although similar measurements (deuterium spectral peaks) have been reported at lower energies by the authors in a previous paper [43] with the same target arrangement, the observation reported here is interesting and novel, in terms of the effect of the prepulse, and more significant in terms of energies observed. The data is certainly worth of publication in some form.

3 experimental spectra are shown in Fig.1. If the nominal conditions in these shots are the same, this points to a mechanism difficult to control in a stable manner, and therefore less significant for applications. In terms of applications and general interest, it is also not expressed clearly what progress a monoenergetic deuterium source could enable, as this is discussed in generic and not very convincing terms. I would also have some reservations with the term quasi-monoenergetic in relation to spectra as shown in 1(f) and ,particularly, 1 (g), which shows a series of peaks over a relatively broad spectral range.

The interpretation of the high-energy signal in the Thomson traces as due to deuterium ions is plausible. However the RCF data interpretation is unclear – how can it be excluded that these are protons at lower energy instead? A discussion of typical proton cut-off and what proton energies would be detected by the layer under discussion would be needed, together with the profiles observed in the precedent RCF layers.

The interpretation of the acceleration as due to Coulomb explosion from a standing wave is an interesting and creative concept, but possibly still not conclusive at this stage.

First of all, it is not explained clearly why the Coulomb explosion should give a directional (or even collimated) beam. There is mention of the bending of the target front due to hole boring as a possible cause, but since the Coulomb explosion process should take place in the underdense plasma in front of it, it is not clear how target denting would affect the divergence of the deuteron ions.

Additionally, the simulation presented does not reflect the experimental conditions closely enough to draw conclusions, beyond a general hint. In the simulations the incidence is normal and therefore the reflected pulse propagates backwards along the incidence direction, while in the experiment the incidence is quite shallow (~40 degrees), so that the relative propagation dynamics (and likely overlap) of the incident and reflected pulse will be quite different. Furthermore, the simulation is run at significantly higher intensity than the experiment as a focal spot 5 times smaller is assumed, so matching between experimental and simulated deuterium energies may be incidental. The authors mention that the smaller focus is used in order to take into account self-focusing in the preplasma, but one could argue that this would be an effect already captured by the PIC code.

There is also some apparent inconsistency between the electron density profile in fig. 3(a) and the electric field lineout in fig3 (e), unless the two refer to very different times? (no significant cavitation is observed in 3(a) close to the target).

Some other minor queries:

- In what sense is E_x in 3(e-g) a mean field ? Is it integrated temporally or spatially or else?
- Is 2.04 ps corresponding to the peak of the pulse incident on the target surface?

Also, a more general comment is that, while the introduction and conclusion are written well, the central part of the manuscript (particularly pages 4-5) is written poorly and difficult to follow.

Reviewer #3

(Remarks to the Author)

The manuscript “Quasi-monoenergetic Deuteron Acceleration via Boosted Coulomb Explosion by Reflected Picosecond Laser Pulse” reports on experimental results of deuteron acceleration driven by picosecond laser pulse. In the experiments, the authors used two laser pulses with a 5 ps delay in between the two pulses, and observed peaked deuteron spectrum at the target front. It is a very interesting topic, however, in my opinion, the manuscript does not justify its publication in Nature Communications, at least under the current version. My concerns are listed below:

- 1) The main experimental finding in this manuscript, is the measured peaked deuteron with energy up to ~50 MeV, and the acceleration process was interpreted by coulomb explosion increased (boosted) by a standing wave induced by the incident picosecond laser pulse. PIC simulations were used to explain the authors’ statement. However, in my opinion, the connection between the experimental data and the simulations are a bit weak, the simulations need to be improved to better support the experimental data.
- 2) Firstly, for the simulation parameters, it is a bit arbitrary. The authors used a three-layered step-like plasma to represent the pre-expanded target. But why they could use such configuration to simulation the target, it is not described in the manuscript. Do they simulate the expansion induced by the pre-pulse? How could they determine such configuration? Could the authors clarify this?
- 3) Secondly, in a previous study done by the same group [Phys. Plasma. 31, 073903 (2024)], they used an almost identical target and obtained a peaked deuteron from the target rear and they used TNSA to explain the results. Those two results are quite similar, the authors need to explain the differences for the mechanisms in a clearer way and illustrate why this time the

dominant mechanism is coulomb explosion. Also, since the target is almost identical, why in the current manuscript, there is no peaked spectrum from the target rear?

4) Enhanced ion acceleration from a standing wave is also reported before [Phys. Rev. Lett. 117, 104801 (2016)], which is quite similar to the experimental results shown in the current manuscript, but explained in a rather different mechanism. Could the authors comment on that? Also, it would be fair to include this paper as a citation and have a clarification in the manuscript for the different acceleration mechanism.

5) There are quite a few missing information for the presented experimental results. For example, the 3 shots showed in Fig. 1e-g, what is the laser and target parameters for those three shots? Are they identical? Fig. 1h showed the measured energy profile by RCF, but is the center of RCF aligned with the target normal at the target front? And does the RCF measured the similar peaked spectrum as observed by the Thomson parabola spectrometer?

6) Few questions for the peaked deuteron beams, is that single shot data, what is the reproducibility for the results? And, the peaked deuteron beam actually presents multi-peaks feature, any explanation for that? Also, the authors mentioned the deuteron has higher energy than protons, could they also plot the proton spectrum at the same shots?

7) The authors stated that the divergence of the high energy deuterons is about 0.03 sr, corresponds to a half angle of 5 degree. I actually have two questions about this. First one, how could they connect the steradian value to degree? Do they assume a axial symmetry for the deuterons? Even though, I am not sure if the numbers are right. Second one, what do they mean about the high energy deuterons? The highest energy? Or a certain energy range in the high energy end with similar divergence? I would ask the authors plotted divergence angle versus the energy from the measurement, to give a better illustration.

8) The authors also try to connect the measured divergence to the simulations. But what is the divergence of the deuteron from the simulations? Would that match the experimental results? Could the authors plot the divergence from the simulations?

9) Few minor issues, in figure 2d, there is a decrease in the resulting deuteron energy, but why does this happened? Could the authors explain it clearly? Also, Fig. 3 shows the E and B fields at different time. But the timing are not the same as in Fig. 2, could the authors clarify this?

10) In Methods, for the laser parameters (presented in the parts of Experiments and Simulations), the numbers need to be more specific. For example, the pulse duration, spot size, does the numbers corresponds to a FWHM value? Also, wavelength of 1.05 micrometer, is that the center wavelength?

Version 1:

Reviewer comments:

Reviewer #1

(Remarks to the Author)

The authors have made significant improvements to the manuscript that substantially strengthens the authors' argument. In particular, the original concerns 1.1 – 1.5 and 1.8 – 1.10 have been addressed well.

1.6: The new simulation at 42° incidence supports the argument well. However, figure 5 (a) is difficult to interpret. The authors should include the incident laser propagation direction and target orientation in the figure, and describe the features they interpret as standing wave (e.g. nodes/antinodes). In addition, the difference in divergence angle between simulation and experiment should be discussed.

1.7: The authors may consider including some details from the response into the manuscript, in particular the calculation of the expansion distance (10 keV deuterons move about 5 μm in 5 ps) and that the acceleration process is not sensitive to the last layer because of its low density.

However, some changes raised additional concerns.

1 Given the new data in table 1 and the response to reviewer comment 2.1, where the authors state that they intended to keep the experimental conditions constant, the vast differences in the deuteron spectra are concerning. Between shot L5168, producing a narrow peak around 20 MeV, and shot L5158, producing a broad peak around 41 MeV, the average laser pulse duration is 1.78 ps vs 1.74 ps. The individual pulses for these shots are also at most 0.15 ps apart. It seems questionable that these fluctuations, or the fluctuations in laser energy, cause these vast spectral differences. The authors explain that four pulses converge simultaneously on the target. Are there any fluctuations in relative beam timing that could significantly change the pulse duration?

2 The diminishing TPIS resolution, as now discussed in the manuscript, should be quantified in the region above 30 MeV to see if it really does broaden the spectra.

3 The authors may wish to consolidate the summary of their previous study (reference 45), which is separately summarized starting in line 56 and also line 116.

Reviewer #3

(Remarks to the Author)

First of all, I would like to thank the authors for taking their efforts to address all the concerns and incorporating appropriate modifications to the manuscript. I recommend its publication in Nature Communications.

One minor comment for the current manuscript, as the authors has replaced “quasi-monoenergetic” with “structured” in the text, I suggest they do the same way for the title and abstract as well.

Dear Reviewers,

Thank you for your thoughtful and constructive comments on our manuscript entitled "Quasi-monoenergetic Deuteron Acceleration via Boosted Coulomb Explosion by Reflected Picosecond Laser Pulse". Your feedback has been invaluable in improving the quality and clarity of our work.

We have carefully revised the manuscript in response to all the comments provided. A detailed point-by-point response is presented below. We believe that our responses address all of your concerns and significantly strengthen the manuscript.

Yours sincerely,
On behalf of the authors

Reviewer #1 (Remarks to the Author):

Quasi-monoenergetic Deuteron Acceleration via Boosted Coulomb Explosion by Reflected Picosecond Laser Pulse

Summary, noteworthy results and significance

The authors describe their experiment of directing a short-pulse laser at an angle onto a D₂O-covered aluminum target. The laser-facing side emitted deuterons with mean energies between 20 and 40 MeV and widths between ~2 and 15 MeV, a significant improvement over typical, exponential laser ion acceleration spectra. The authors explain this atypical, quasi-monoenergetic spectrum by a new laser-ion acceleration mechanism, a ‘boosted Coulomb explosion’ caused by the reflected laser beam. Simulations from a corresponding 2D PIC simulation are presented to support this conclusion.

The work is an extension of the author’s previous study (Wei, T. et al, Physics of Plasmas 31(7), 073903 (2024)). The main improvement is an extension of their results to higher deuteron beam energies by accelerating deuterons off the front (laser-facing) side instead of the back. Other studies have limited themselves to deuteron energies <10 MeV, whereas this study went up to 40 MeV, albeit with larger energy spread. Therefore, it is a valuable addition to the body of literature.

Response to reviewer:

We sincerely thank the reviewer for the thorough summary of our work and for recognizing the significance of our results. We greatly appreciate the comments raised by the reviewer. Our point-by-point responses are as follows.

Comment 1.1:

On p.3, it is difficult to follow what ‘rear’ and ‘front’ side mean and where the D₂O layer is. The authors may consider using ‘laser-facing’ instead.

Response to 1.1:

We thank the reviewer for this suggestion. To improve clarity, we have consistently replaced the term “front” with “laser-facing” throughout the manuscript. Additionally, we have added a subfigure in the blank space of Fig. 1(e) to clearly indicate the position of the D₂O layer and the laser-facing side for better visualization.

Comment 1.2:

The figure 1 (a)-(e) labels seem shifted in the text.

Response to 1.2:

The figure labels have been corrected in the revised manuscript.

Comment 1.3:

The specific laser parameters should be included for each shot shown in figure 1 (e)-(g).

Response to 1.3:

At Line 247, we change the sentence of describing laser basic parameters to

“The LFEX laser provides three shots per day and four pulses of H1–H4 with a center wavelength of 1.05 μm are simultaneously delivered for each shot. The focal spot diameters of the four pulses are 50 μm for H1, H3, and H4 and 30 μm for H2. The laser pulse energy and duration of each shot used in the experiments are listed in Table 1”

We also added the more detailed laser parameters for every shot in the newly added Table 1 at Experiments part.

Comment 1.4:

The authors present RCF to show the spatial beam profile. It is not clear from the text whether the RCF is also sensitive to protons, which could obscure the deuteron beam profile.

Response to 1.4:

We have verified that proton contributions do not affect the interpretation of the presented beam profile using the PHITS particle transport simulation code. The proton energy spectra measured by the TPIS (shown in the added Fig. 2) indicate the maximum energy is limited only to 23 MeV. In contrast, the calculation using the PHITS particle transport simulation code [51] shows that the minimum energies of protons and deuterons reaching to the RCF layer under discussion are 24.6 MeV and 33.0 MeV, respectively. Since the proton maximum energy is below the reaching energy of this RCF layer, the observed signals are attributed only to deuterons with energies higher than 33.0 MeV.

At Line 104, we added the following clarification.

“By calculating the particle transport with the PHITS Monte Carlo simulation code [51], the minimum energies of the protons and deuterons reaching to the RCF layer presented in Figs. 1(h) and (j) through other layers is 24.6 MeV and 33.0 MeV, respectively. Because the maximum energy of the protons is 23 MeV as shown in Fig. 2, the signals in the RCF originate only from to the deuterons. ”

Comment 1.5:

Figure 2 (b) would benefit from including the experimental deuteron spectrum to show a match

between simulation and experiment.

Response to 1.5:

We have added the experimental deuteron spectrum from shot L5165 and L5158 in addition to the simulation results in Fig. 3(b).

At Line 131, we added the following sentence to describe this change.

“The energy spectra measured in the shot L5165 and L5158 are also included in Fig. 3(b) to illustrate the agreement between simulation and experiment results. ”

Comment 1.6:

Figure 2 reveals that the simulation used a laser pulse impinging on the target at 90° . However, the experiment shown in figure 4 shows a 41.8° incidence. The TNSA mechanism is generally robust with respect to this angle; however, the authors later argue for the generation of a standing wave between incident and reflecting laser. This would require the wave vectors sum to zero, which is unlikely at this angle. The authors should repeat their PIC simulation with this angle, or otherwise demonstrate that a standing wave can form under these conditions.

Response to 1.6:

Thank you for pointing out this important issue. We have performed an additional PIC simulation with a laser incident angle of 42° , which is close to the experimental angle of 41.8° . As shown in Fig.5(a), a standing wave structure forms in the region where the incident and reflected pulses overlap. The standing wave forms along the normal direction of the target surface, which ensures that the Boosted Coulomb Explosion mechanism remains effective. In general, the standing wave forms along the normal direction independent of the incident angle of the laser.

At Line 194, we added the following paragraph to make this clear to readers.

“In the above simulations, the angle of the incident laser has been assumed to be 0° from the normal direction of the target surface. Because the incident angle in the experiment is 41.8° , we perform an additional simulation where the incident angle of 42° is assumed. Figure 5(a) presents that a standing wave still forms in the overlap region of the incident and reflected pulses and the quasi-monoenergy deuterons are accelerated up to approximately 80 MeV along the normal direction with a open divergence angle of smaller than 24° shown in Fig. 5(b).”

At Line 300, we added the following paragraph of the set-up.

“The simulation box for the oblique incidence in Fig. 5 is in the region of from -200 to 50 μm

for the x-axis and from -100 to 300 μm for the y-axis with a mesh size of 50 nm. The laser spot (FWHM) is set as 25 μm with the same laser energy. The target set-up is the same with that for the normal incidence. ”

Comment 1.7:

The simulation includes densities for the solid target and deuteron plasma, which should be explained or justified

Response to 1.7:

The densities in the simulation were chosen to approximate the experimental conditions after pre-pulse expansion while maintaining computational feasibility. The target was modeled as a three-layer structure. The solid region was set to 80 n_c , representing aluminum solid target where protons were used instead of Al ions to reduce computational cost. In the laser-facing side of this region, an exponential deuteron plasma profile with a peak density of 30 n_c was located, corresponding to the fully ionized D₂O layer ($\sim 33n_c$). To simulate the pre-pulse with an energy of $\sim 5\%$ of the main pulse (~ 30 J) reaching 5 ps before the main pulse, we set the deuteron layer expand partially. By assuming a deuteron temperature of ~ 10 keV, the expansion distance is estimated to be ~ 5 μm in 5 ps. To represent the most expanded front, we added a 6.5 μm layer with a density of 0.018 n_c . The acceleration process is not highly sensitive to this layer’s length because of its low density. Oxygen ions are not set in deuteron plasma for reducing computation complexity. This configuration captures the essential density gradient in experiments.

At Line 277, we added the following sentences to make these clear to readers.

“As shown in Fig. 3(a) protons with a density of 80 n_c are set in the solid Al target region from 0 to 5 μm for the x-axis. The protons are used instead of Al ions because of computational efficiency. By considering a laser pre-pulse, low-density deuteron plasma with a density of 0.018 n_c is set in the region from -10 to -3.5 μm for the x-axis to present the pre-pulse expansion, which is estimated to have a temperature of 10 keV over 5 ps. High-density deuterons is set in the region from -3.5 to 0 μm for the x-axis to represent the fully ionized D₂O layer ($\sim 33 n_c$). This deuteron layer has an exponential distribution of $30 n_c e^{-2|x|}$ at the laser-facing side of the proton layer. Note that oxygen ions are not set in the deuteron plasma because of computational efficiency.”

Comment 1.8:

On p.4, the authors make the interesting observation that for pulse durations >1.5 ps, the reduced laser intensity cannot penetrate the ionized deuterons. How does this depend on the

chosen densities, and are there other reasonable choices?

Response to 1.8:

The ability of the laser to penetrate the ionized deuteron layer depends strongly on the pre-plasma density and its scale length. In our simulations, the pre-expanded D₂O layer was modeled with an exponential distribution and a scale length of a few micrometers. In the condition that the total laser pulse energy is fixed, when the pulse duration exceeds ~1.5 ps, the peak intensity and normalized vector potential a_0 decrease, reducing the relativistic skin depth and limiting the length of effective penetration. If the pre-plasma density is lower or the scale length is shorter, the penetration condition would relax, allowing longer pulses to maintain efficient electron evacuation and Coulomb explosion field strength. Conversely, higher densities or thicker layers would favor shorter pulses.

At Line 140, we have revised the manuscript as follows to clarify this.

“The calculated result [Fig. 3(d)] shows that the maximum peak energy of the accelerated deuterons is obtained in the case of a pulse duration of 1.5 ps, which is nearly equal to the typical pulse width of the LFEX laser. If we use laser pulses longer than 1.5 ps, the peak intensity and normalized vector potential a_0 decrease ($a_0 \propto 1/\sqrt{\tau}$) under the condition that the total laser pulse energy is fixed. As a_0 decreases, the penetration depth of a laser pulse in a relativistic plasma (relativistic skin depth) shortens and the amplitude of the reflected laser to form a standing wave decreases. These effects reduce the strength of the boosted Coulomb explosion field to accelerate deuterons. As shown in Fig. 3(d), when pulse widths are longer than 1.5 ps, the peak energies of the accelerated deuterons are lower than that at 1.5 ps by approximately 5-8 MeV. The energy spectrum of the accelerated deuterons depends on the density and the scale length of the pre-plasma. In the present condition, the pre-expanded D₂O layer has an exponential profile with a few-micrometer scale length. If the pre-plasma has a thin scale length or low density, a long laser pulse penetrates easily the pre-plasma and accelerates effectively deuterons. Conversely, when the pre-plasma has a thick scale length or a high density, a shorter laser pulse is suitable for effective deuteron acceleration.”

Comment 1.9:

On p.5, the authors argue that the high intensity laser pulse ejects most electrons at the laser-facing side, leaving the ions, which then primarily interact with themselves, leading to a Coulomb explosion that produces the quasi-monoenergetic deuterons. The argument would be significantly strengthened if the authors could separately calculate the electric field generated by the electron density shown in figure 3 (a) and demonstrate that it is insignificant.

Response to 1.9:

We sincerely thank the reviewer for this insightful suggestion. Fig. 4(a) shows the electron

density distribution n_e , from which the electric field generated by electrons can be estimated using Poisson's equation with the charge density $\rho = -en_e$. At the laser-facing side ($x < 0$) the maximum strength of the electric field generated by the electrons is estimated to be on the order of 10^9 V/m. In contrast, the acceleration field experienced by the deuterons at the laser-facing side is on the order of teravolts per meter (10^{12} V/m), indicating that the Coulomb explosion field is the dominant mechanism for deuteron acceleration.

At Line 179, we have added the following sentence to clarify this point for readers.

“The electric field generated by electrons at the laser-facing side could be estimated from the electron density distribution shown in Fig. 4(a). The estimated field is on the order of 10^9 V/m, which is much lower than the Coulomb explosion field strength of TV/m.”

Comment 1.10:

On p.7-8, some more experimental details should be given, in particular the temperature of the Al target during deposition and the specific laser parameters used.

Response to 1.10:

The room temperature of the experimental room where other various laser systems have been also located has been controlled to keep a constant temperature of 25°C. The evaporation of D₂O molecular occurred without any heat sources inside the vacuum chamber. Thus, the Al target should remain at the room temperature during deposition even though there should be slight variations.

By considering your former comment 1.3 together, the specific laser parameters for all shots have already been listed in the added Table 1 at Experiments part.

At Line 273, we add the sentence

“The Al targets are kept at near room temperature (25°C) during the deposition process”.

Finally, we would like to thank the reviewer for the kind and detailed comments. We have improved the manuscript a lot.

Reviewer #2 (Remarks to the Author):

The manuscript by Wei et al, presents some interesting observations of narrow band deuterium spectra from LFEX irradiations of Deuterium-coated substrates.

The narrow band, high energy features, observed in the target front direction appear only when a prepulse is added to the interaction, suggesting a mechanism taking place in a pre-expanded deuterium-containing plasma. The author suggest a mechanism based on Coulomb explosion in a region of the front plasma where the electrons are evacuated by a standing wave arising from interference of the incident laser and the reflected laser. The interpretation is supported by 2D-PIC simulation showing a similar acceleration pattern.

Although similar measurements (deuterium spectral peaks) have been reported at lower energies by the authors in a previous paper [43] with the same target arrangement, the observation reported here is interesting and novel, in terms of the effect of the prepulse, and more significant in terms of energies observed. The data is certainly worth of publication in some form.

Response to reviewer:

Thank you for carefully reading our manuscript and positive summary for it. We appreciate the useful comments, the point-by-point responses are as follows.

Comment 2.1:

3 experimental spectra are shown in Fig.1. If the nominal conditions in these shots are the same, this points to a mechanism difficult to control in a stable manner, and therefore less significant for applications.

Response to 2.1:

Although we tried to keep conditions as constant, the laser variations contributed to the observed spectral differences. We use high power laser with pulse energy of 600 J, and four pulses are focused on the target. Please see the added Table 1 at Experiment part, the laser pulse energy of one pulse varied from 118 J to 192 J, and the pulse duration also varied in the range of 1.44 – 1.92 ps in FWHM shot-by-shot. These fluctuations cause the unstable of the spectra of the accelerated ions.

Comment 2.2:

In terms of applications and general interest, it is also not expressed clearly what progress a monoenergetic deuterium source could enable, as this is discussed in generic and not very convincing terms.

Response to 2.2:

One of the main purposes of the deuteron acceleration is generation of laser-driven neutron sources. It has been known that when quasi-monochromatic deuteron beams are used with the well-designed collimator and beamlines, neutrons with constrained energy distribution can be provided. This contributes to various neutron applications. Another specific application is production of medical radioisotopes. At present, medical radioisotopes such as $^{99}\text{Mo}/^{98\text{m}}\text{Tc}$ have been generated using large-scale system such as nuclear reactors and accelerators. A compact laser production system located inside hospitals can provide quickly medical radioisotopes when a medical doctor requires it for a patient. In particular, deuteron beams have been studied for medical radioisotopes such as ^{64}Cu and ^{177}Lu . Another application is the study of the DD and DT nuclear reactions using high-power lasers. Mono-energy deuterons are useful for the measurements of the cross section at a specific energy and control of these reactions. Furthermore, as a conceptual extension, similar setups could potentially generate monoenergetic tritium beams, enabling studies of TT reactions. Such experiments are challenging with conventional accelerators due to severe radiation contamination issues.

At Line 47, we have added the following sentences to clarify this point for readers.

“For example, monoenergetic deuterons via high power laser can be used for neutron sources with selective energy using the secondary reactions such as the $\text{d}+^9\text{Be}$ reaction,[29] ,the study of $\text{d}+\text{d}$ and $\text{d}+\text{t}$ nuclear fusion [30, 31] , and production of medical radioisotopes [32]”

Comment 2.3:

I would also have some reservations with the term quasi-monoenergetic in relation to spectra as shown in 1(f) and ,particularly, 1 (g), which shows a series of peaks over a relatively broad spectral range.

Response to 2.3:

We appreciate the reviewer’s concern regarding the use of the term “quasi-monoenergetic” to describe the spectra shown in Fig. 1(f) and, particularly, Fig. 1(g), which indeed exhibit a series of peaks over a relatively broad energy range compared to Fig. 1(e).

We believe that two main factors contribute to these spectral structures:

1. Detector resolution: Our TPIS diagnostic system provides high energy resolution (<0.2 MeV) in the 3–30 MeV range. However, at higher energies (>30 MeV), the resolution decreases, which may broaden the observed spectral features.
2. Physical mechanism: Even accounting for resolution limitations, the peak structures remain

evident, especially in Fig. 1(e), which lies within the high-resolution range. As noted by Reviewer #3, similar multi-peak ion spectra have been predicted in [Phys. Rev. Lett. 117, 104801 (2016)], where a femtosecond laser-induced standing wave creates a spatially modulated electron distribution, leading to a structured sheath field and discrete ion energy bands. In our case, although a picosecond laser is used, we consider it plausible that during the early stage (tens of femtoseconds), the transient electron modulation from the standing wave may imprint a multi-peak structure on the deuterons before the main Coulomb explosion acceleration occurs.

We modified the manuscript as follows.

To better reflect the nature of the observed spectra and avoid potential misinterpretation, we have replaced the term “quasi-monoenergetic” with “structured” in the description of the experimental results.

At Line 100, we added

“Note that the TPISs used in the experiments have an energy resolution better than 0.2 MeV in the 3–30 MeV range. At higher energies (>30 MeV), the resolution decreases, which may broaden the observed spectral features.”

At Line 190, we added the following paragraph to discuss the possible physical mechanism behind the multi-peak structures.

“Note that multi-peak structures are observed in all three experimental shots (Fig. 1(e)–(g)). During the early stage when the standing wave forms at tens of femtoseconds, electron modulation may induce a multi-peak structure on the deuterons [47] before the main Coulomb explosion acceleration.”

Comment 2.4:

The interpretation of the high-energy signal in the Thomson traces as due to deuterium ions is plausible. However the RCF data interpretation is unclear – how can it be excluded that these are protons at lower energy instead? A discussion of typical proton cut-off and what proton energies would be detected by the layer under discussion would be needed, together with the profiles observed in the precedent RCF layers.

Response to 2.4:

To address this, we have added the proton energy spectra measured at the laser-facing side using the TPIS in Fig. 2. The proton maximum energy is approximately 23 MeV. By calculation of the particle transport with the PHITS simulation code, the minimum energy of protons and

deuterons reaching on this RCF layer through other layers are 24.6 MeV and 33.0 MeV, respectively. Since the measured proton maximum energy is lower than this threshold energy reaching to this RCF layer, we identified the observed signals in this RCF layer to originate only from deuterons with energies higher than 33.0 MeV.

At Line 104, we added the following clarification.

“By calculating the particle transport with the PHITS Monte Carlo simulation code[51], the minimum energies of the protons and deuterons reaching to the RCF layer presented in Figs. 1(h) and (i) through other layers is 24.6 MeV and 33.0 MeV, respectively. Because the maximum energy of the protons is 23 MeV as shown in Fig. 2, the signals in the RCF originate only from to the deuterons.”

Comment 2.5:

The interpretation of the acceleration as due to Coulomb explosion from a standing wave is an interesting and creative concept, but possibly still not conclusive at this stage.

First of all, it is not explained clearly why the Coulomb explosion should give a directional (or even collimated) beam. There is mention of the bending of the target front due to hole boring as a possible cause, but since the Coulomb explosion process should take place in the underdense plasma in front of it, it is not clear how target denting would affect the divergence of the deuteron ions. Additionally, the simulation presented does not reflect the experimental conditions closely enough to draw conclusions, beyond a general hint. In the simulations the incidence is normal and therefore the reflected pulse propagates backwards along the incidence direction, while in the experiment the incidence is quite shallow (~40 degrees), so that the relative propagation dynamics (and likely overlap) of the incident and reflected pulse will be quite different.

Response to 2.5:

We have conducted an additional PIC simulation with a 42° incidence angle, as shown in Fig. 5. The results confirm that a standing wave still forms in the overlap region of the incident and reflected pulses along the normal direction of the target surface and that deuterons are accelerated to several tens of MeV along the normal direction.

The standing wave leads to localized electron evacuation, whereas electrons at other directions remains around the target, and thus a strong electrostatic field is generated along the normal direction. This accelerates deuterons preferentially along the target normal. The resulting angular distribution, shown in Fig. 5(b), is consistent with the experimental RCF observations. Furthermore, the standing wave is generated along the normal direction independent of the

incidence angle of the laser.

At Line 194, we added the following paragraph to make this clear to readers.

“In the above simulations, the angle of the incident laser has been assumed to be 0° from the normal direction of the target surface. Because the incident angle in the experiment is 41.8° , we perform an additional simulation where the incident angle of 42° is assumed. Figure 5(a) presents that a standing wave still forms in the overlap region of the incident and reflected pulses and the quasi-monoenergy deuterons are accelerated up to approximately 80 MeV along the normal direction with a open divergence angle of smaller than 24° shown in Fig. 5(b). The standing wave leads to localized electron evacuation, and thus a strong electrostatic field is generated along the normal direction. This accelerates deuterons preferentially along the normal direction.”

At Line 300, we added the following paragraph of the set-up.

“The simulation box for the oblique incidence in Fig. 5 is in the region of from -200 to 50 μm for the x-axis and from -100 to 300 μm for the y-axis with a mesh size of 50 nm. The laser spot (FWHM) is set as 25 μm with the same laser energy. The target set-up is the same with that for the normal incidence.”

Comment 2.6:

Furthermore, the simulation is run at significantly higher intensity than the experiment as a focal spot 5 times smaller is assumed, so matching between experimental and simulated deuterium energies may be incidental. The authors mention that the smaller focus is used in order to take into account self-focusing in the preplasma, but one could argue that this would be an effect already captured by the PIC code.

Response to 2.6:

As pointed out by the reviewer, the self-focusing effects were captured by the PIC code. In our simulation, however, we also reduced the focal spot size from 50 μm to 10 μm to manage computational complexity while preserving the essential physics of the interaction. The total laser energy has been kept constant, resulting in a higher input intensity compared to the experiment intensity.

While this intensity difference may affect certain aspects of the interaction, the Coulomb explosion field is primarily governed by the number of evacuated electrons, rather than the laser intensity itself. The increased intensity allows the main pulse to penetrate deeper into the preplasma, shifting the position of the reflection surface to a higher electron density. In our

simulation, the reflected surface is located at a density of approximately $n_c\sqrt{1 + a_0^2/2}$, where $a_0 \approx 0.85\sqrt{I_{18}}\lambda_{\mu m}$. This corresponds to a density roughly 4.5 times higher. However, since the focal spot size is 5 times smaller in the horizontal direction, the total number of evacuated electrons remains comparable to the experimental case. As a result, the strength of the Coulomb explosion field is almost preserved, and the simulation remains representative of the physical mechanism observed in the experiment.

At Line 288, we have added the following sentences to clarify.

“The laser spot size is set to 10 μm in FWHM instead of the size in the experiments of 50 μm to account for self-focusing effects and to reduce computational complexity. The laser intensity of the simulation increases from the experiment intensity of $10^{19}\text{W}/\text{cm}^2$ to approximately $2.5 \times 10^{20}\text{W}/\text{cm}^2$. This higher intensity allows deeper penetration of the main pulse into the pre-plasma, but the reflected surface is shifted to a position of a higher electron density of $n_c\sqrt{1 + a_0^2/2}$ which is approximately 4.5 times higher than that in the experiments. Since the focal spot size is 5 times smaller in the horizontal direction, the total number of evacuated electrons remains comparable to the experimental case. As a result, the total number of evacuated electrons in the simulation is comparable to that in the experiments. This leads that the present simulation giving the Coulomb explosion field strength similar to the experimental condition.”

Comment 2.7:

There is also some apparent inconsistency between the electron density profile in fig. 3(a) and the electric field lineout in fig3 (e), unless the two refer to very different times? (no significant cavitation is observed in 3(a) close to the target).

Response to 2.7:

Fig. 4(a) in the revised manuscript shows the electron density profile at 1.08 ps, corresponding to an early stage when the standing wave is just forming before the deuteron acceleration. In contrast, Fig. 4(e) presents the deuteron energy distribution and electric field lineout after deuteron acceleration has occurred.

The cavitation observed in Fig. 4(e) refers to regions where deuterons have been evacuated due to acceleration. Unlike deuterons, electrons can continuously emerge from the solid target region, so such cavitation is not expected in the electron density profile near the target.

To avoid confusion, we have added a time stamp to Fig. 4(a) in the revised manuscript. Additionally, we have included a time reference in the figure description to clearly indicate that Fig. 4(a) corresponds to an early stage (1.08 ps) when the standing wave is forming.

Comment 2.8:

Some other minor queries:

- In what sense is E_x in 3(e-g) a mean field? Is it integrated temporally or spatially or else?
- Is 2.04 ps corresponding to the peak of the pulse incident on the target surface?

Response to 2.8:

We thank the reviewer for these two insightful questions and apologize for the lack of clarity in our original description.

Regarding the first point, the “mean field” E_x shown in Fig. 4(e–g) refers to the spatial mean value of the electric field in the central region ($-2.5 \mu\text{m}$ to $2.5 \mu\text{m}$). The acceleration field is superimposed with the laser field, making it difficult to distinguish visually. Therefore, we use the spatial average to better illustrate the underlying acceleration field.

As for the second point, the time stamps such as 2.04 ps in Fig. 4(b) correspond to the simulation time. In the simulation, the laser pulse is set from the left boundary with its peak at 1 ps. It takes approximately 0.67 ps for the pulse to reach the target surface, so the peak arrives at around 1.67 ps. Thus, 2.04 ps represents a time shortly after the peak has interacted with the target.

To clarify these points for readers, we have made the following modifications to the manuscript.

At Line 167, we have changed the phrase “the mean value of E_x field” to “the spatial mean value of E_x field”.

At Line 161, we added “The time stamps in these figures show the simulation time. The laser pulse is set from the left boundary ($-200 \mu\text{m}$) with a pulse width of 1.5 ps and a peak at 1 ps. This pulse reaches the surface of the target at approximately 0.67 ps later, and the peak of the pulse reaches this surface at approximately 1.67 ps.”

Comment 2.9

Also, a more general comment is that, while the introduction and conclusion are written well, the central part of the manuscript (particularly pages 4-5) is written poorly and difficult to follow.

Response to 2.9:

We added many explanations in the Result and Method, and we have thoroughly revised the central part of the paper to improve readability and logical flow. Finally, we again thank the

reviewer for many professional and positive comments. The revised manuscript has been much improved from the previous version.

Reviewer #3 (Remarks to the Author):

The manuscript “Quasi-monoenergetic Deuteron Acceleration via Boosted Coulomb Explosion by Reflected Picosecond Laser Pulse” reports on experimental results of deuteron acceleration driven by picosecond laser pulse. In the experiments, the authors used two laser pulses with a 5 ps delay in between the two pulses, and observed peaked deuteron spectrum at the target front. It is a very interesting topic, however, in my opinion, the manuscript does not justify its publication in Nature Communications, at least under the current version. My concerns are listed below.

Response to reviewer:

We thank the reviewer for the brief conclusion of our work and the professional comments. The point-by-point responses are as follows.

Comment 3.1:

The main experimental finding in this manuscript, is the measured peaked deuteron with energy up to ~50 MeV, and the acceleration process was interpreted by coulomb explosion increased (boosted) by a standing wave induced by the incident picosecond laser pulse. PIC simulations were used to explain the authors’ statement. However, in my opinion, the connection between the experimental data and the simulations are a bit weak, the simulations need to be improved to better support the experimental data.

Response to 3.1:

Thank you for your valuable comments regarding the connection between the experimental data and the simulations. We performed new PIC simulations with a 42° incidence angle to better match the experimental configuration and provide improved insight into the interaction.

Comment 3.2:

Firstly, for the simulation parameters, it is a bit arbitrary. The authors used a three-layered step-like plasma to represent the pre-expanded target. But why they could use such configuration to simulation the target, it is not described in the manuscript. Do they simulate the expansion induced by the pre-pulse? How could they determine such configuration? Could the authors clarify this?

Response to 3.2:

The three-layer step-like plasma configuration was introduced to approximate the pre-expanded target profile created by the pre-pulse while keeping the simulation computationally feasible. We estimated the density profile based on the pre-pulse energy (~5% of the main pulse of

≈ 30 J), its timing (5 ps before the main pulse), and previous studies on similar targets. The solid region was set to $80 n_c$ to represent aluminum, and protons were used instead of Al ions to reduce computational cost. In front of this region, an exponential deuteron plasma profile with a peak density of $30 n_c$ was introduced, corresponding to the fully ionized D_2O layer ($\sim 33 n_c$). Considering the pre-pulse heating, the layer would partially expand, assuming a deuteron temperature of ~ 10 keV, the expansion distance is estimated to be $\sim 5 \mu\text{m}$ in 5 ps. To represent the most expanded front, we added a thin $6.5 \mu\text{m}$ layer with a density of $0.018 n_c$. This simplified configuration captures the essential density gradient in experiments without performing a full hydrodynamic simulation, and sensitivity tests confirmed that the acceleration process is not highly dependent on the exact length of the low-density layer.

At Line 277, we add the following sentences to make these clear to readers.

“As shown in Fig. 3(a) protons with a density of $80 n_c$ are set in the solid Al target region from 0 to $5 \mu\text{m}$ for the x-axis. The protons are used instead of Al ions because of computational efficiency. By considering a laser pre-pulse, low-density deuteron plasma with a density of $0.018 n_c$ is set in the region from -10 to $-3.5 \mu\text{m}$ for the x-axis to present the pre-pulse expansion, which is estimated to have a temperature of 10 keV over 5 ps. High-density deuterons is set in the region from -3.5 to $0 \mu\text{m}$ for the x-axis to represent the fully ionized D_2O layer ($\sim 33 n_c$). This deuteron layer has an exponential distribution of $30 n_c e^{-2|x|}$ at the laser-facing side of the proton layer. Note that oxygen ions are not set in the deuteron plasma because of computational efficiency.”

Comment 3.3:

Secondly, in a previous study done by the same group [Phys. Plasma. 31, 073903 (2024)], they used an almost identical target and obtained a peaked deuteron from the target rear and they used TNSA to explain the results. Those two results are quite similar, the authors need to explain the differences for the mechanisms in a clearer way and illustrate why this time the dominant mechanism is coulomb explosion. Also, since the target is almost identical, why in the current manuscript, there is no peaked spectrum from the target rear?

Response to 3.3:

We greatly appreciate the reviewer’s careful reading of our previous work [Phys. Plasmas 31, 073903 (2024)]. First, we now consistently use “laser-facing side” instead of “front side” and “rear side” for clarity.

In the previous work, the heavy water (D_2O) layer was located in the “rare side”. In contrast, in the present study, the heavy water layer was located in the “laser-facing side”. This difference

leads to a completely different acceleration mechanism.

In the previous study, the observed peaked deuteron spectrum from the rear side was explained by the TNSA mechanism. The ultra-thin heavy water layer (tens of nanometers) allowed most deuterons to be accelerated early from the rear side, while protons were accelerated later. Some high-speed and low-speed protons compressed the deuterons from both sides, forming a peaked component.

In the present study, the laser is incident on the heavy water side, and a pre-pulse (~ 5 ps before the main pulse) is introduced to expand the ultra-thin heavy water layer. A standing wave is formed by the incident and reflected pulses, which evacuates electrons from the laser-facing side along the normal direction of the target surface. This results in suppresses the formation of a strong sheath field for TNSA. Instead, the deuterons are accelerated by Coulomb explosion along the normal direction on the laser-facing side.

At Line 116, we have added the following paragraph to clarify these differences.

“In the previous work [45] we accelerated quasi-monoenergetic deuterons near 11 MeV using a picosecond laser pulse with an energy of approximately 600 J. From an ultra thin D₂O layer (tens of nanometers) at the rear side of the target (no D₂O layer in the laser-facing side), deuterons are accelerated early by the TNSA mechanism, and protons are also accelerated later. High and low velocity protons compress the deuteron distribution from both velocity regions in the momentum space, and thereby a component of quasi-monoenergetic deuterons is formed. In contrast, in the present study, the deuterons are accelerated from the laser-facing side of the target and their energies are higher than those of the simultaneously accelerated protons. These facts indicate that the quasi-monoenergetic deuterons are accelerated by a mechanism different from the TNSA in the previous study. Note that the detection threshold of the TPIS is approximately 3 MeV for deuterons.”

Comment 3.4:

Enhanced ion acceleration from a standing wave is also reported before [Phys. Rev. Lett. 117, 104801 (2016)], which is quite similar to the experimental results shown in the current manuscript, but explained in a rather different mechanism. Could the authors comment on that? Also, it would be fair to include this paper as a citation and have a clarification in the manuscript for the different acceleration mechanism.

Response to 3.4:

We sincerely thank the reviewer for pointing out this important reference [Phys. Rev. Lett. 117, 104801 (2016)], which we had previously overlooked. That study made a significant

contribution by demonstrating that a femtosecond laser-induced standing wave can generate a spatially modulated electron distribution, resulting in a structured sheath field and multi-peak ion spectra.

In our experiment, the standing wave is formed by a picosecond laser pulse, which operates on a much longer timescale. This enables substantial electron evacuation from the laser-facing side, making Coulomb explosion the dominant acceleration mechanism rather than sheath acceleration. Nevertheless, our experimental spectra do exhibit a series of peaks similar to those predicted in the cited work. We consider it plausible that, during the early stage (tens of femtoseconds) when the standing wave forms, electron modulation may imprint a multi-peak structure on the deuterons before the main Coulomb explosion occurs.

Thus, our results can be viewed as complementary to the mechanism proposed in the cited paper, extending the concept of standing-wave-enhanced ion acceleration to a different temporal regime and physical condition.

To clarify this relationship, we have cited the reference in the Introduction and added a discussion in the Results section highlighting both the similarities and differences.

At Line 71, we have added the following sentence.

“In addition, it has been theoretically proposed that protons could be accelerated up to 100 MeV through a longitudinal charge-separation field generated by chirped standing waves formed by a fs laser pulse reflected by a high-density mirror located behind a target [47]”.

To improve the flow of the Introduction, we have added the word “picosecond” before “laser pulse” at the next paragraph of it.

At Line 190, we added the following paragraph.

“Note that multi-peak structures are observed in all three experimental shots [Fig. 1(e)–(g)]. During the early stage when the standing wave forms at tens of femtoseconds, electron modulation may induce a multi-peak structure on the deuterons [47] before the main Coulomb explosion acceleration.”

Comment 3.5:

There are quite a few missing information for the presented experimental results. For example, the 3 shots showed in Fig. 1e-g, what is the laser and target parameters for those three shots? Are they identical? Fig. 1h showed the measured energy profile by RCF, but is the center of RCF aligned with the target normal at the target front? And does the RCF measured the similar

peaked spectrum as observed by the Thomson parabola spectrometer?

Response to 3.5:

To clarify the experimental conditions, we have added Table 1 in the Methods section, which lists the laser parameters for each shot shown in Fig. 1(e–g). While the targets were fabricated using the same process to ensure consistency, the laser energy and pulse duration varied slightly between shots. These variations are the main reason for the differences observed in the deuteron spectra.

The RCF stack was placed parallel to the target surface but offset by approximately 1 cm from the target centerline to avoid interference with the incoming laser beam. As a result, the deuteron pattern appears shifted on the RCF image.

Due to the relatively coarse energy resolution of the RCF compared to the Thomson parabola ion spectrometer (TPIS), and the strong proton background in the lower energy layers, we were unable to resolve a similarly peaked deuteron spectrum from the RCF data.

To clarify these points for readers, we have revised the manuscript as follows.

At Line 263, after the RCF setup description, we added:

“The RCF stack was placed parallel to the target surface but offset by approximately 1 cm from the centerline to avoid interference with the laser beam.”

At Line 109, we added:

“It is difficult to evaluate the deuteron energy spectrum lower than 33 MeV from the RCF stacks, because both protons and deuterons make signals in the lower energy RCF layers.”

Comment 3.6:

Few questions for the peaked deuteron beams, is that single shot data, what is the reproducibility for the results? And, the peaked deuteron beam actually presents multi-peaks feature, any explanation for that?

Response to 3.6:

Each energy spectrum was obtained from a single-shot measurement. To assess reproducibility, we conducted six LFEX shots over two experimental days (three shots per day). Among these, two shots were performed without a pre-pulse, three shots with a pre-pulse were measured using TPIS, and one shot with a pre-pulse was measured using RCF stacks. All three TPIS shots with a pre-pulse exhibited peaked deuteron spectra, and the RCF measurement also confirmed the presence of high-energy deuterons consistent with the TPIS results. These results indicated that

the generation of high-energy deuterons is highly reproducible when a pre-pulse is applied.

Regarding the multi-peak feature, we point out that each spectrum was obtained from a single-shot measurement and not the result of integral multiple shots. As discussed in our response to Comment 3.4, we attribute the multi-peak structure to early-stage electron modulation induced by a standing wave, which may imprint discrete energy features on the deuterons prior to the main Coulomb explosion acceleration.

At Line 254, we have added the following sentence to clarify these points for readers.

“For all the shots with pre-pulse, we have high-energy components of deuterons at laser-facing side which indicate a high reproducibility.”

At line 97, we added “Each energy spectrum is obtained from a single laser shot without accumulation.”

Comment 3.7:

Also, the authors mentioned the deuteron has higher energy than protons, could they also plot the proton spectrum at the same shots?

Response to 3.7:

We added the proton energy spectra measured by TPIS in Fig. 2.

Comment 3.8:

The authors stated that the divergence of the high energy deuterons is about 0.03 sr, corresponds to a half angle of 5 degree. I actually have two questions about this. First one, how could they connect the steradian value to degree? Do they assume a axial symmetry for the deuterons? Even though, I am not sure if the numbers are right.

Response to 3.8:

The divergence of the high-energy deuterons was estimated based on the area of the high-energy spot observed on the RCF ($\pi \times 2.2\text{mm} \times 4.4\text{mm} \approx 30.41\text{mm}^2$) and the distance from the target (31 mm), yielding a solid angle of $\Omega = 30.41/31^2 \approx 0.03$ sr. To convert this solid angle into a half-angle in degrees, we assumed an axially symmetric conical emission. Under this assumption, the half-angle θ can be derived from the relation: $\Omega = 2\pi(1 - \cos\theta)$, Solving this gives $\theta \approx 5.6^\circ$. We apologize for the discrepancy in the original manuscript and have corrected the value from 5° to 5.6° .

At Line 113, we have added the calculation and explicitly to state the assumption.

“By considering the distance of the RCF from the target (31 mm), the divergence of the high-energy deuterons is estimated as $\Omega=\pi\times 2.23\times 4.4/31^2\approx 0.03$ sr. Assuming an axially symmetric conical distribution, the half angle $\theta\approx 5.6^\circ$ is derived from $\Omega=2\pi(1-\cos\theta)$.”

Comment 3.9:

The authors also try to connect the measured divergence to the simulations. But what is the divergence of the deuteron from the simulations? Would that match the experimental results? Could the authors plot the divergence from the simulations?

Response to 3.9:

To address this point, we performed additional PIC simulation with a 42° incidence angle (Fig. 5), we extracted the angular distribution of the accelerated deuterons. In the added simulations, we find that the standing wave is always formed along the target normal direction and deuterons are accelerated along the normal direction. The obtained open divergence angle of approximately 24° is larger than the experimental result.

At Line 194, we added the following paragraph to make this clear to readers.

“In the above simulations, the angle of the incident laser has been assumed to be 0° from the normal direction of the target surface. Because the incident angle in the experiment is 41.8° , we perform an additional simulation where the incident angle of 42° is assumed. Figure 5(a) presents that a standing wave still forms in the overlap region of the incident and reflected pulses and the quasi-monoenergy deuterons are accelerated up to approximately 80 MeV along the normal direction with a open divergence angle of smaller than 24° shown in Fig. 5(b).”

At Line 300, we added the following paragraph of the set-up.

“The simulation box for the oblique incidence in Fig. 5 is in the region of from -200 to 50 μm for the x-axis and from -100 to 300 μm for the y-axis with a mesh size of 50 nm. The laser spot (FWHM) is set as 25 μm with the same laser energy. The target set-up is the same with that for the normal incidence. ”

Comment 3.10:

Few minor issues, in figure 2d, there is a decrease in the resulting deuteron energy, but why does this happened? Could the authors explain it clearly?

Response to 3.10:

The decrease in deuteron energy for longer pulse durations (Fig. 3(d)) occurs because the total laser energy was kept constant while varying the pulse duration. As the pulse becomes longer,

its peak intensity and normalized vector potential a_0 decrease ($a_0 \propto 1/\sqrt{\tau}$). A lower a_0 reduces the relativistic skin depth and weakens the ability of the incident and reflected pulses to form a strong standing wave and expel electrons from the target front side. This diminishes the boosted Coulomb explosion field responsible for deuteron acceleration. Although a longer pulse slightly extends the acceleration time, the reduction in field strength dominates, resulting in a net decrease in peak energy beyond the optimal duration (~ 1.5 ps).

At Line 140, we have revised the manuscript as follows to clarify this.

“The calculated result [Fig. 3(d)] shows that the maximum peak energy of the accelerated deuterons is obtained in the case of a pulse duration of 1.5 ps, which is nearly equal to the typical pulse width of the LFEX laser. If we use laser pulses longer than 1.5 ps, the peak intensity and normalized vector potential a_0 decrease ($a_0 \propto 1/\sqrt{\tau}$) under the condition that the total laser pulse energy is fixed. As a_0 decreases, the penetration depth of a laser pulse in a relativistic plasma (relativistic skin depth) shortens and the amplitude of the reflected laser to form a standing wave decreases. These effects reduce the strength of the boosted Coulomb explosion field to accelerate deuterons. As shown in Fig. 3(d), when pulse widths are longer than 1.5 ps, the peak energies of the accelerated deuterons are lower than that at 1.5 ps by approximately 5-8 MeV. The energy spectrum of the accelerated deuterons depends on the density and the scale length of the pre-plasma. In the present condition, the pre-expanded D₂O layer has an exponential profile with a few-micrometer scale length. If the pre-plasma has a thin scale length or low density, a long laser pulse penetrates easily the pre-plasma and accelerates effectively deuterons. Conversely, when the pre-plasma has a thick scale length or a high density, a shorter laser pulse is suitable for effective deuteron acceleration.”

Comment 3.11:

Also, Fig. 3 shows the E and B fields at different time. But the timing are not the same as in Fig. 2, could the authors clarify this?

Response to 3.11:

In Fig. 4, the snapshots at 2.04 ps, 2.64 ps, and 3.24 ps were selected to best illustrate the evolution of the standing wave structure. In contrast, Fig. 3 originally showed energy spectra at 2.04 ps, 2.64 ps, and 4.44 ps, with 4.44 ps corresponding to the final stage of the simulation (up to 4.5 ps), representing the last stable deuteron spectrum.

To improve consistency between the two figures and facilitate clearer comparison between the field evolution and the corresponding energy spectra, we have added the spectrum at 3.24 ps to Fig. 4 in the revised manuscript.

Comment 3.12:

In Methods, for the laser parameters (presented in the parts of Experiments and Simulations), the numbers need to be more specific. For example, the pulse duration, spot size, does the numbers corresponds to a FWHM value? Also, wavelength of 1.05 micrometer, is that the center wavelength?

Response to 3.12:

To address this, we have added an additional Table 1 in the Methods section summarizing all laser parameters used in the experiments. The pulse duration and focal spot size correspond to FWHM values, and the wavelength of 1.05 μm refers to the center wavelength of the laser.

At Line 247, we have revised the description of the laser parameters as follows.

“The LFEX laser provides three shots per day and four pulses of H1-H4 with a center wavelength of 1.05 μm are simultaneously delivered for each shot. The focal spot diameters of the four pulses are 50 μm for H1, H3, and H4 and 30 μm for H2. The laser pulse energy and duration of each shot used in the experiments are listed in Table 1”

We have also specified that the pulse duration and focal spot size in the simulation parameters correspond to FWHM values.

Furthermore, at Line 288, we added the sentence: “The wavelength is 1.05 μm consistent with experiments.”

Finally, we would like to thank again the reviewer #3 for his kind and detailed questions and suggestions. We believe that the revised manuscript was much improved, and we understand physics behind our results more deeply.